# Humoral and Mucosal Antibody Response to RSV Structural Proteins in RSV-Infected Adult Hematopoietic Cell Transplant (HCT) Recipients

**DOI:** 10.3390/v13060991

**Published:** 2021-05-26

**Authors:** Xunyan Ye, Obinna P. Iwuchukwu, Vasanthi Avadhanula, Letisha O. Aideyan, Trevor J. McBride, David M. Henke, Kirtida D. Patel, Felipe-Andres Piedra, Laura S. Angelo, Dimpy P. Shah, Roy F. Chemaly, Pedro A. Piedra

**Affiliations:** 1Department of Molecular Virology & Microbiology, Baylor College of Medicine, Houston, TX 77030, USA; xunyan.ye@bcm.edu (X.Y.); Obinna.Iwuchukwu@bcm.edu (O.P.I.); avadhanu@bcm.edu (V.A.); aideyan@bcm.edu (L.O.A.); Trevor.McBride@bcm.edu (T.J.M.); David.Henke@bcm.edu (D.M.H.); kpatel@bcm.edu (K.D.P.); Felipe-Andres.Piedra@bcm.edu (F.-A.P.); Laura.Angelo@bcm.edu (L.S.A.); 2Department of Epidemiology and Biostatistics, The University of Texas Health Science Center at San Antonio, San Antonio, TX 78229, USA; ShahDP@uthscsa.edu; 3Departments of Infectious Diseases, Infection Control and Employee Health, The University of Texas MD Anderson Cancer Center, Houston, TX 77030, USA; rfchemaly@mdanderson.org; 4Department of Pediatrics, Baylor College of Medicine, Houston, TX 77030, USA

**Keywords:** respiratory syncytial virus, RSV, humoral antibody, mucosal antibody, hematopoietic cell transplant adults

## Abstract

Respiratory syncytial virus (RSV) is an important cause of lower respiratory tract infection in infants, the elderly, and immunocompromised patients. RSV antibodies play a role in preventing reinfection and in clearance of RSV, but data regarding the levels of viral protein-specific antibodies elicited and their contribution to patient recovery from RSV-induced disease are limited. We prospectively enrolled a cohort of RSV-infected adult hematopoietic cell transplant (HCT) recipients (*n* = 40). Serum and nasal-wash samples were obtained at enrollment (acute samples) and convalescence (convalescent samples). We measured (1) humoral IgG and mucosal IgA binding antibody levels to multiple RSV proteins (F, G, N, P, and M2-1) by Western blot (WB); (2) neutralizing antibody (Nt Ab) titers by microneutralization assay; and (3) palivizumab-like antibody (PLA) concentrations by an ELISA-based competitive binding assay developed in the lab. Finally, we tested for correlations between protein-specific antibody levels and duration of viral shedding (normal: cleared in <14 days and delayed: cleared ≥14 days), as well as RSV/A and RSV/B subtypes. Convalescent sera from HCT recipients had significantly higher levels of anti-RSV antibodies to all 5 RSV structural proteins assayed (G, F, N, P, M2-1), higher Nt Abs to both RSV subtypes, and higher serum PLAs than at enrollment. Significantly higher levels of mucosal antibodies to 3 RSV structural proteins (G, N, and M2-1) were observed in the convalescent nasal wash versus acute nasal wash. Normal viral clearance group had significantly higher levels of serum IgG antibodies to F, N, and P viral proteins, higher Nt Ab to both RSV subtypes, and higher PLA, as well as higher levels of mucosal IgA antibodies to G and M2-1 viral proteins, and higher Nt Ab to both RSV subtypes compared to delayed viral clearance group. Normal RSV clearance was associated with higher IgG serum antibody levels to F and P viral proteins, and PLAs in convalescent serum (*p* < 0.05). Finally, overall antibody levels in RSV/A- and/B-infected HCT recipients were not significantly different. In summary, specific humoral and mucosal RSV antibodies are associated with viral clearance in HCT recipients naturally infected with RSV. In contrast to the humoral response, the F surface glycoprotein was not a major target of mucosal immunity. Our findings have implications for antigen selection in the development of RSV vaccines.

## 1. Introduction

Respiratory syncytial virus (RSV) is a global human pathogen that can cause severe respiratory disease in infants, the elderly, and the immunocompromised host [1,2,3]. Despite the enormous burden of RSV disease and decades of research, no vaccine or antiviral drug therapy has been licensed, although there are close to 40 vaccine candidates in different phases of development [4]. Palivizumab, the only FDA-approved therapeutic for the prevention of RSV disease, is a monoclonal antibody (mAb) targeting RSV fusion (F) glycoprotein and is used in a select group of high-risk infants [5]. Ascertaining immune correlates of protection can assist in the selection of promising vaccine candidates for late phase development [6,7,8].

The RSV genome is about 15.2 kb long and encodes 11 proteins including two nonstructural (NS1 and NS2) proteins followed, in gene order, by nucleocapsid (N), phosphoprotein (P), matrix (M), small hydrophobic (SH), attachment (G) surface glycoprotein, fusion (F) surface glycoprotein, M2 (M2-1 and M2-2) protein, and the RNA-dependent RNA polymerase (L). The G glycoprotein plays a role in host cell attachment and immune evasion [9]. The F protein allows the virion membrane to fuse with the target cell membrane [10], and the SH protein inhibits apoptosis [11,12] and acts as a viroporin [13]. The G and F surface glycoproteins are the only known RSV proteins that induce neutralizing antibodies after natural infection. RSV P, N, and L proteins interact to form a polymerase complex for viral RNA transcription and replication [14,15]. The M2 protein consists of the M2-1 protein and the M2-2 protein. M2-1 is an essential cofactor of the viral RNA polymerase complex and functions as a transcriptional processivity and anti-termination factor, while M2-2 protein is a regulatory factor involved in the balance between RNA replication and transcription [16]. Measurements of antibody levels to the viral surface glycoproteins and internal viral structural proteins have been used to study the immune response following vaccination or infection with wild type RSV [17,18,19].

The generally accepted principle is that both mucosal and humoral antibodies are important for protection from RSV infection. In infants and young children, serum neutralizing antibody levels correlate robustly with protection against severe RSV-induced lower respiratory tract illness (LRTI) [8,20]. Intravenous immunoglobulin containing high levels of RSV-specific neutralizing antibodies and the monoclonal antibody palivizumab that targets site II on the F glycoprotein protects high-risk infants against RSV-LRTI hospitalization [21]. In adults experimentally challenged with RSV, levels of neutralizing antibodies in upper respiratory tract secretions appear to be a better immune correlate of protection than levels of serum neutralizing antibodies [22,23]. Recently, Bagga et al. demonstrated that higher pre-existing RSV-specific serum and nasal antibodies predicted lower infectivity in a human challenge model, but showed no significant protective effect once infection occurred [24]. Durable RSV-specific mucosal IgA responses may be more protective than humoral antibody [25].

We recently described a cohort of adult hematopoietic stem cell transplant (HCT) recipients who were naturally infected with RSV [26]. Approximately 50% of these adults cleared RSV within 2 weeks of virus identification (normal viral clearance) and the other half experienced delayed viral clearance. Significant rises in both serum neutralizing antibody titers and palivizumab-like antibody concentration was associated with normal viral clearance [26]. In the present study, we further evaluated the mucosal and humoral RSV protein-specific immune responses to RSV infection in this HCT cohort and identified additional immune correlates of viral clearance.

## 2. Methods

### 2.1. Study Design

We previously reported on the serum neutralizing antibody, palivizumab-like antibody, and antigenic-site specific antibody responses in 40 RSV-infected HCT adults [26,27,28]. We now expand our findings to compare the RSV protein-specific antibody responses detected in the serum (humoral) to those generated in the upper respiratory tract (mucosal). From January 2012 to April 2015, sera were collected from all patients at enrollment (acute) and 14–60 days after hospitalization (convalescent). Nasal-wash samples were collected at enrollment (acute), day 7 (±1), day 14 (±1), and between day 21 and day 28 (±1). Medical records were reviewed to collect demographic and clinical data. 

### 2.2. RSV Detection

Real time, reverse transcription polymerase chain reaction (RT-PCR) was performed on the nasal washes as previously described for the detection of RSV/A and RSV/B from this cohort [29].

### 2.3. RSV-Specific Microneutralization (MN) Assay

Serum and nasal-wash samples were analyzed for neutralizing antibodies (Nt Ab) against RSV/A/Tracy and RSV/B/18537 in HEp-2 cells using a qualified microneutralization (MN) assay as previously described [18,30,31]. Neutralizing antibody titers were defined as the final dilution at which there was a 50% reduction in viral cytopathic effect (CPE). Any sample resulting in a titer less than the lower limit of detection (LLoD: 2.5 log2) was assigned a value of 2 log2.

### 2.4. Western Blot (WB)

The Western blot (WB) assay was used for the detection and semi-quantification of serum IgG and secretory IgA to the surface glycoproteins of RSV (attachment (G) and fusion (F)), and to the internal structural proteins (nucleoprotein (N), phosphoprotein (P) and matrix 2 (M2-1)) as previously described [18,19,20]. Sucrose purified RSV/A/Bernett (spRSV) was used as the antigen and was purified using established methods [17,32]. Briefly, spRSV was denatured and added to the wells of an SDS-PAGE polyacrylamide gel. The proteins were separated for 17 h, and then electrophorectically transferred to a 0.45 µm nitrocellulose membrane for 1 h followed by overnight blocking with 3% gelatin at 37 °C. In each assay, up to 10 membranes, each placed in a PR150 Deca-Probe 10-channel manifold (Hoefer Scientific Instruments: HSI Hoefer), were available for testing. The first membrane was used to confirm the banding patterns using protein-specific polyclonal and monoclonal antibodies. The subsequent nine membranes were used to test the serum and nasal-wash samples. The outer wells of each membrane were not used followed by Precision Plus Protein Dual Color low molecular weight Standards (Bio-Rad) that bracketed up to six samples. The serum samples of an individual were included in the same membrane. Similarly, the nasal washes of an individual were included in a single membrane. Serum and nasal-wash samples (1:50 dilution) that were heat inactivated in a 56 °C water bath for 30 min were added, followed by a 2 h incubation at 37 °C with gentle rocking. The wells were washed and horseradish peroxidase (HRP) conjugated secondary antibodies (anti-IgG at dilution 1/2000 or anti-IgA at dilution 1/2000) was added to all the wells and allowed to incubate at 37 °C for 2 h, followed by seven-minute incubation with substrate 3,3′,5,5′-tetramethylbenzidine (TMB) with enhancement (Kirkegaard and Perry Labs). For an assay to meet acceptance criteria, the positive control membrane had to demonstrate banding patterns specific to the G, F, N, P, and M2-1 proteins and a negative PBS control. Antibody responses to the G, F, N, P, M2-1 viral proteins were recognized by bands corresponding to molecular weights in the 70–90 kDa, 46–50 kDa, 39–41 kDa, 33–35 kDa, and 25 kDa, respectively. A new protein-specific response was recognized by one or more new bands or an increase in the intensity of one or more bands between the acute and convalescent samples. Semi-quantitation was achieved by use of a visual log developed to gauge the relative intensity of the band. The band intensity scores assigned were 0 (no band), 5 (faint band), 10 (light band with thickness of ~1 mm), 20 (more intense band with thickness of ~2 mm), 30 (intense band with thickness of >2 mm), and 40 (most intense band with thickness of >3 mm).

### 2.5. Palivizumab Competitive Antibody (PCA) Assay

PCA assay, as previously described [27], was used to measure the concentrations of palivizumab-like antibodies (PLA, an antigenic site II specific antibody) that compete with palivizumab for binding to antigenic site II of the fusion protein of RSV in sera and nasal-wash samples. The source of the fusion protein was from sucrose purified RSV/A/Bernett (spRSV) that was coated onto a 96-well plate for 18 h at 4 °C. One hundred µL of palivizumab at concentration of 1.25 µg/mL in 5% milk was added in duplicate followed by 2-fold serial dilutions (12,500 ng/mL to 24.41 ng/mL) for generating a standard curve on each plate. Wells containing biotinylated palivizumab without samples served as positive controls representing maximum binding. Wells containing 5% milk instead of samples and wells containing sera without biotinylated palivizumab served as negative controls. A four-parameter logistic (4PL) regression model was used to calculate the PLA concentrations (µg/mL) based on the dynamic range of the standard curve by interpolating the concentration of the standards that corresponds to the absorbance value at which the test sample resulted in 50% inhibition. The LLoD was 1 µg/mL. Samples with concentration below the LLoD were assigned a value of 0.5 µg/mL.

### 2.6. Statistical Analysis

For demographic characteristics and clinical outcomes, continuous variables were compared using two-sample *t*-test or Mann–Whitney U-test. Categorical variables were compared using the chi-squared test or Fisher’s exact test. Paired *t*-tests were used to determine whether the means of log transformed PLA concentrations (log2 µg/mL) differed significantly between acute and convalescent samples. Mann–Whitney U-test was used to determine whether the mean ranks of Nt Ab log2 titers or WB band density scores differed significantly between acute and convalescent samples, between HCT adults who cleared RSV from the upper respiratory tract (URT) in <14 (normal viral clearance) and ≥14 days (delayed viral clearance), and between RSV/A- and RSV/B-infected HCT adults. Two-sample *t*-test was used to determine whether the means of log transformed PLA concentrations differed significantly from normal viral clearance versus delayed viral clearance HCT adults, and between RSV/A- and RSV/B-infected adults. To determine clinical and laboratory factors associated with viral clearance time in HCT recipients, a binary regression model was created. Backwards selection of variables, using a likelihood ratio test with a *p*-value of 0.10 cutoff, was used to reduce the numbers of factors. Antibodies measured by WB were coded as ordinal variables while Nt Ab and PLA were treated as continuous variables. It was first employed to determine which clinical covariates best predicted the dependent variable, normal viral clearance in HCT recipients. The logistic regression model with the significant clinical covariate was then used to determine which antibodies best predicted the dependent variable. Antibodies were tested independently of one another. Statistical significance was indicated for *p*-values < 0.05. Statistical analyses were performed using the SPSS 22 (IBM Corp., Armonk, NY, USA).

## 3. Results

### 3.1. Study Subjects

Clinical and demographic characteristics at enrollment of the 40 RSV-infected HCT adults were previously described [26]. Age, gender, race/ethnicity, absolute neutrophil counts (ANC), absolute lymphocyte counts (ALC), and median time from HCT to RSV infection were comparable between the groups when stratified by duration of RSV shedding or by RSV infection subtype. The only significant difference observed was that recipients who shed RSV from the URT for >14 days were more likely to have received an allogeneic stem cell transplant compared to recipients who shed virus for less than 14 days (18/20 versus 11/20; *p* = 0.02).

### 3.2. Comparison of Humoral and Mucosal RSV Antibody Level in Acute and Convalescent Samples

Neutralizing, binding and competitive antibodies specific to RSV were evaluated in the serum and respiratory secretion at enrollment and 14–60 days after hospitalization by microneutralization, WB, and PCA assays, respectively. In the convalescent serum samples (Table 1), the median titers to RSV/A and RSV/B significantly increased by approximately eight-fold compared to the titers in the acute serum samples. A significant increase in serum IgG antibodies to the two major surface glycoproteins (G and F) and to the internal proteins (N, P and M2-1) was detected. PLA activity also increased significantly. In contrast, in the convalescent respiratory samples, IgA antibodies against the G, N, and M2-1 proteins were the only significant increases detected. Increases in functional antibody activity, as measured by the microneutralization assay or PCA assay, were not detected in the respiratory secretions collected during convalescence.

### 3.3. Comparison of Humoral and Mucosal Anti-RSV Antibody Levels in HCT Recipients Who Shed RSV for <14 Versus ≥14 Days

We previously reported on this cohort that HCT adults with normal viral clearance mounted significant increases in PLA, F antigenic site-specific antibodies, and Nt Ab activity compared to those with delayed viral clearance [26,27,29]. Thus, we expected to see significant increases in humoral and mucosal RSV-specific responses to RSV structural proteins in the group with normal viral clearance. We compared the normal and delayed viral clearance groups at enrollment (acute) and convalescent time points for humoral and mucosal anti-RSV-specific antibody responses (Table 2). At enrollment, no significant differences were detected in the humoral and mucosal RSV-specific antibody levels between the two groups. At the convalescent time point, HCT adults in the normal viral clearance group had significantly greater RSV-specific antibody levels in the serum and respiratory secretions compared to those in the delayed viral clearance group. In the serum, the significant increases were detected in Nt Ab, IgG antibodies to the F, N, and P viral proteins, and to PLA in the normal viral clearance group. By contrast, the increased antibody activity observed in the respiratory secretions were Nt Ab, and IgA antibodies to the G and M2-1 viral proteins in the normal viral clearance group.

### 3.4. Association of Humoral and Mucosal RSV Antibody Level with Virus Resolution in RSV-Infected HCT Recipients

We further analyzed the humoral and mucosal RSV antibody responses to determine whether a particular clinical variable and/or a particular antibody response would be a statistically significant predictor of virus resolution in HCT recipients. Using binary logistic regression, we analyzed all 8 clinical variables: injury severity score (ISS), treatment arm (oral, aerosolized, control, not applicable), age, gender, race (white, black, Hispanic, Asian), body mass index (BMI), type of transplant (autologous, allogeneic), and concurrent infections at onset of RSV. The analysis showed that patient transplant type was the only clinical variable which showed association with viral clearance (data not shown). Therefore, it was the only covariate used when testing RSV antibody level association with normal viral clearance. As shown in Table 3, significant association of normal RSV clearance was observed for three RSV antibodies in the convalescent serum: IgG anti-F, IgG anti-P, and PLA antibodies (*p* < 0.05). Increases in serum IgG binding antibodies to F and P viral proteins, and PLA were associated with an increased likelihood of normal virus clearance, with point estimates for the ORs of 0.89, 0.86, and 0.64, respectively. A significant association with viral clearance was not observed with any of the binding and functional mucosal antibody responses.

### 3.5. Comparison of Humoral and Mucosal RSV Antibody Level between RSV/A- and RSV/B-Infected HCT Recipients

Infection caused by RSV/A and RSV/B could result in subtype-specific variation in RSV-specific antibody responses especially towards the G surface glycoprotein. We compared RSV/A- and RSV/B-infected groups at enrollment and convalescent time points for humoral and mucosal RSV-specific antibody responses. No significant differences in the levels of serum or mucosal antibodies were observed between the RSV/A- and RSV/B-infected groups for neutralizing, binding, and competitive antibodies specific to RSV (Table 4).

## 4. Discussion

The magnitude of humoral and mucosal antibody responses to various RSV antigens, including RSV-specific F, G, N, P, and M2-1 and to antigenic sites such as site II of the F protein, and the magnitude of neutralizing antibody responses after natural infection in patients have been partially characterized in human RSV challenge and RSV vaccine studies. In this report, we extend our previous findings on the humoral immune response in RSV-infected HCT adults [26,27,29]. Using the Western blot assay to complement the neutralizing antibody and PLA responses we observed major differences in the antigenic targets of mucosal IgA compared to that of humoral IgG RSV-specific antibody responses. The IgA response in the nasal wash was primarily to the G and not the F surface glycoprotein based on the Western blot assay. This finding is supported by both the lack of PLA activity, which is F protein-specific, and the absence of a significant increase in neutralizing antibody activity in the nasal-wash samples after RSV infection. The PLA is an F site II specific antibody response and the microneutralization assay that we used primarily detects neutralizing antibody activity that is generated by the F and not the G protein [33,34]. On the other hand, the IgG response in serum was directed to both the F and G glycoproteins, and resulted in significant increases in PLA and neutralizing antibody activity after RSV infection. This dichotomy in antigen-specific antibody responses between mucosal and humoral immunity likely reflects the major viral antigenic targets being detected in these immunologically distinct compartments during an RSV infection in HCT adults.

Our results may have implications for other populations that are naturally infected with RSV or vaccinated with a live attenuated RSV vaccine. Differences in the immunologic repertoire of the upper respiratory tract and serum has been observed previously in human RSV challenge studies. A lack of correlation between RSV-specific pre-existing mucosal IgA and serum neutralizing antibody titers within individual volunteers suggest that they likely offer protection by independent mechanisms [25]. Nasal neutralizing antibody has been shown to be a better correlate of protection than serum neutralizing antibody [22]. Additionally, resistance to intranasal RSV inoculation in adult volunteers correlated with neutralizing antibody titers in nasal secretions and to a lesser extent in serum [23]. Interestingly, reformatting palivizumab and motavizumab from IgG isotype to human IgA isotype impaired their protective efficacy against RSV infection in HEp-2 cells and in mice [35]. Taken together the humoral and mucosal antibody responses appear distinct and play different role in protection against infection and disease.

Our data showed significant rises in all 8 humoral antibodies (IgG G, F, N, P, M2-1, RSV/A Nt Ab, RSV/B Nt Ab, and PLA) and 3 mucosal antibodies (IgA G, IgA N, and IgA M2-1) measured in convalescent sera. Our data strongly support the ongoing use of Nt Ab, IgG anti-F, and PLA as a readout for natural infection that could be used for vaccine development. Our findings are also consistent with another study [36] suggesting that vaccination should also aim to evoke a strong mucosal immune response. Mucosal IgA may have a more important role in protection of the upper respiratory tract than humoral IgG [37]. Furthermore, in post RSV-infected HCT recipients who shed virus for <14 days, our study showed a significant increase in levels of all but two (IgG G and IgG M2-1) humoral antibodies assayed, and a significant increase in mucosal IgA anti-G, IgA anti-M2-1, and RSV Nt Ab levels, suggesting those Abs contributed to virus clearance and patient recovery. Our data are consistent with prior studies, each involving experimental infection of healthy adults, showing association between reduced RSV replication in the upper respiratory tract and higher levels of (1) pre-inoculation humoral RSV Nt Ab, and (2) RSV-specific mucosal IgA activity [24,25,38].

RSV A and B subtypes are estimated to have diverged over 350 years ago [39]. The two subtypes differ most greatly in the G gene, while other genes are well conserved. We examined humoral and mucosal immune responses to RSV/A and RSV/B, and found that both subtypes elicit similar responses in HCT recipients. In apparent contrast, other researchers have reported distinct patterns of innate immune responses in terms of cytokine/chemokine induction by different clinical RSV/A isolates [40].

Our study has some limitations. The small sample size of naturally infected HCT recipients (*n* = 40) is not representative of adults in the general population; however, it does represent a group that is highly susceptible to the severe diseases of RSV infection and the study will contribute to the knowledge of vaccine development in this vulnerable group of people. The Western blot assay used for detecting the antibodies primarily bind linear epitopes on the RSV structural proteins and may not detect the conformationally sensitive anti-RSV antibodies present in the serum and nasal-wash samples. However, we have previously reported on F-specific conformation-dependent competitive antibodies in this HCT population [26,27,28]. The microneutralization assay used a continuous cell line as its matrix to primarily detect neutralizing antibody activities generated by the F instead of the G protein. In addition, there was no investigation of the cell-mediated immune response in this cohort, an important immune parameter for investigating viral clearance.

In conclusion, this study characterizes and compares humoral and mucosal immune responses to multiple RSV structural proteins in naturally infected HCT recipients. Our data shows that increases in specific humoral and mucosal anti-RSV antibodies are associated with viral clearance; and we find no evidence for a viral subtype-dependent adaptive immune response in adults with reinfection. These data can serve as a resource for the evaluation of candidate RSV vaccines for adult HCT recipients.

## Figures and Tables

**Table 1 viruses-13-00991-t001:** Comparison of Humoral and Mucosal RSV Antibody Level in Acute and Convalescent Samples.

Sample Types	Antibody	Acute Samples (*n* = 40)	Convalescent Samples (*n* = 40)	*p*-Value ^2^
Serum	RSV/A Nt Ab	6.8 (5.5–8.4) ^1^	9.5 (7.6–11.0)	**<0.001**
	RSV/B Nt Ab	7.0 (6.0–8.9)	9.8 (7.0–12.5)	**<0.001**
	IgG G	5.0 (1.3–17.5)	10.0 (5.0–20.0)	**0.018**
	IgG F	0.0 (0.0–0.0)	2.5 (0.0–10.0)	**0.004**
	IgG N	0.0 (0.0–0.0)	5.0 (0.0–17.5)	**0.002**
	IgG P	0.0 (0.0–5.0)	5.0 (0.0–10.0)	**0.007**
	IgG M2-1	0.0 (0.0–5.0)	5.0 (0.0–17.5)	**0.001**
	PLA	2.6 (0.5–182.2)	10.5 (0.5–768.0)	**<0.001**
Nasal Wash	RSV/A Nt Ab	2.0 (2.0–2.4)	2.0 (2.0–2.0)	0.237
	RSV/B Nt Ab	2.0 (2.0–3.0)	2.0 (2.0–2.0)	0.526
	IgA G	2.5 (0.0–10.0)	7.5 (5.0–30.0)	**0.001**
	IgA F	0.0 (0.0–5.0)	2.5 (0.0–5.0)	0.251
	IgA N	0.0 (0.0–5.0)	2.5 (0.0–10.0)	**0.007**
	IgA P	0.0 (0.0–5.0)	0.0 (0.0–5.0)	0.729
	IgA M2-1	0.0 (0.0–0.0)	0.0 (0.0–0.0)	**0.014**
	PLA	0.5 (0.5–0.5)	0.5 (0.5–0.5)	NA

^1^ Median (IQR) of log2 titers, band intensity scores and µg/mL were used for RSV Nt Ab, WB Ab, and PLA, respectively. ^2^ Mann–Whitney U-test (2 related samples approach) was used for difference in mean ranks of Nt Ab log2 titers and WB band density scores; Paired *t*-test was used for difference in means of log transformed PLA concentrations. NA = Not applicable. The bold values mean that the *p*-values are statistically significant at 0.05 level. The serum RSV Nt Ab and serum PLA data were published as geometric mean (95% confidence interval) in our previous publications [27].

**Table 2 viruses-13-00991-t002:** Comparison of Humoral and Mucosal RSV Antibody Level between RSV-Infected HCT Recipients Who Shed RSV < or ≥14 Days.

RSV Ab Type	Virus Shedding Period (Days)		*p*-Values ^2^
	<14 (*n* = 20)	≥14 (*n* = 20)	
Humoral Ab in acute serum			
RSV/A Nt Ab	7.3 (5.5–8.5) ^1^	6.3 (5.1–8.0)	0.447
RSV/B Nt Ab	6.8 (6.0–9.9)	7.0 (5.6–8.0)	0.337
IgG G	5.0 (1.3–17.5)	7.5 (1.3–17.5)	0.625
IgG F	0.0 (0.0–0.0)	0.0 (0.0–3.8)	0.725
IgG N	0.0 (0.0–3.8)	0.0 (0.0–0.0)	0.236
IgG P	0.0 (0.0–5.0)	0.0 (0.0–5.0)	0.947
IgG M2-1	0.0 (0.0–5.0)	0.0 (0.0–5.0)	0.479
PLA	2.9 (0.5–182.2)	2.5 (0.5–10.4)	0.139
Humoral Ab in convalescent serum			
RSV/A Nt Ab	10.8 (8.5–12.8)	8.8 (6.5–9.9)	**0.014**
RSV/B Nt Ab	11.5 (8.8–13.5)	8.5 (7.0–10.4)	**0.007**
IgG G	10.0 (6.3–20.0)	7.5 (5.0–20.0)	0.361
IgG F	10.0 (0.0–20.0)	0.0 (0.0–5.0)	**0.003**
IgG N	10.0 (1.3–20.0)	0.0 (0.0–5.0)	**0.005**
IgG P	7.5 (5.0–20.0)	0.0 (0.0–5.0)	**0.001**
IgG M2-1	10.0 (5.0–20.0)	5.0 (0.0–5.0)	0.053
PLA	78.7 (1.5–768.0)	9.6 (0.5–117.0)	**<0.001**
Mucosal Ab in acute nasal wash			
RSV/A Nt Ab	2.0 (2.0–3.0)	2.0 (2.0–2.0)	0.160
RSV/B Nt Ab	2.0 (2.0–5.3)	2.0 (2.0–2.4)	0.089
IgA G	5.0 (0.0–10.0)	0.0 (0.0–10.0)	0.621
IgA F	0.0 (0.0–8.8)	0.0 (0.0–3.8)	0.155
IgA N	0.0 (0.0–0.0)	0.0 (0.0–5.0)	0.116
IgA P	0.0 (0.0–5.0)	0.0 (0.0–5.0)	0.491
IgA M2-1	0.0 (0.0–0.0)	0.0 (0.0–0.0)	0.317
PLA	0.5 (0.5–0.5)	0.5 (0.5–0.5)	NA
Mucosal Ab in convalescent nasal wash			
RSV/A Nt Ab	2.0 (2.0–2.5)	2.0 (2.0–2.0)	**0.047**
RSV/B Nt Ab	2.0 (2.0–4.5)	2.0 (2.0–2.0)	**0.009**
IgA G	25.0 (5.0–40.0)	5.0 (0.0–20.0)	**0.027**
IgA F	2.5 (0.0–10.0)	2.5 (0.0–5.0)	0.703
IgA N	5.0 (0.0–10.0)	0.0 (0.0–8.8)	0.221
IgA P	5.0 (0.0–5.0)	0.0 (0.0–5.0)	0.156
IgA M2-1	0.0 (0.0–5.0)	0.0 (0.0–0.0)	**0.004**
PLA	0.5 (0.5–0.5)	0.5 (0.5–0.5)	NA

^1^ Median (IQR) of log2 titers and band intensity scores were shown for RSV Nt Ab and WB Ab; Median (Range) of µg/mL was shown for PLA. ^2^ Mann–Whitney U-test was used for difference in mean ranks of RSV Nt Ab titers and WB band density scores; Two-sample *t*-test was used for difference in means of log transformed PLA concentrations. IQR = interquartile range; NA = Not applicable. The bold values mean that the *p*-values are statistically significant at 0.05 level. The serum RSV Nt Ab and serum PLA data were published as geometric mean (95% confidence interval) in our previous publications [27].

**Table 3 viruses-13-00991-t003:** Odds Ratios (ORs) for Association of Specific RSV Antibody with Virus Resolution in RSV-Infected HCT Recipients.

RSV Ab Type	OR (95% CI)	*p*-Values ^1^
Humoral Ab in acute serum (*n* = 40)		
RSV/A Nt Ab	0.85 (0.60–1.21)	0.376
RSV/B Nt Ab	0.88 (0.64–1.21)	0.428
IgG G	1.00 (0.94–1.10)	0.633
IgG F	1.10 (0.97–1.29)	0.124
IgG N	0.97 (0.86–1.10)	0.642
IgG P	1.04 (0.92–1.17)	0.567
IgG M2-1	1.02 (0.92–1.12)	0.768
PLA	0.86 (0.54–1.36)	0.516
Humoral Ab in convalescent serum (*n* = 40)		
RSV/A Nt Ab	0.82 (0.59–1.13)	0.227
RSV/B Nt Ab	0.79 (0.59–1.07)	0.134
IgG G	0.99 (0.93–1.05)	0.704
IgG F	0.89 (0.79–0.99)	**0.032**
IgG N	0.91 (0.83–1.00)	0.054
IgG P	0.86 (0.75–0.98)	**0.023**
IgG M2-1	0.97 (0.90–1.04)	0.363
PLA	0.64 (0.45–0.92)	**0.016**
Mucosal Ab in acute nasal wash (*n* = 40)		
RSV/A Nt Ab	0.67 (0.31–1.45)	0.308
RSV/B Nt Ab	0.57 (0.27–1.19)	0.133
IgA G	0.99 (0.93–1.06)	0.780
IgA F	0.90 (0.77–1.05)	0.164
IgA N	1.11 (0.92–1.34)	0.270
IgA P	0.94 (0.84–1.06)	0.332
IgA M2-1	0.11 (0.00–NA)	1.000
PLA	NA	NA
Mucosal Ab in convalescent nasal wash (*n* = 40)		
RSV/A Nt Ab	0.29 (0.05–1.91)	0.200
RSV/B Nt Ab	0.45 (0.18–1.14)	0.091
IgA G	0.96 (0.91–1.01)	0.095
IgA F	1.00 (0.91–1.10)	0.979
IgA N	0.96 (0.88–1.05)	0.373
IgA P	0.87 (0.72–1.05)	0.147
IgA M2-1	0.02 (0.00–NA)	0.999
PLA	NA	NA

^1^ Binary logistic regression was used for the data analysis. Abbreviations: CI = confidence interval; OR = odds ratio; NA = not applicable. *p*-value = a score test to predict whether or not an independent variable would be a statistically significant predictor in the model. The bold values mean that the *p*-values are statistically significant at 0.05 level.

**Table 4 viruses-13-00991-t004:** Comparison of Humoral and Mucosal RSV Antibody Level between RSV/A- and RSV/B-Infected Patients.

RSV Ab Type	RSV Subtype		*p*-Values ^2^
	RSV/A (*n* = 22)	RSV/B (*n* = 18)	
Humoral Ab in acute serum			
RSV/A Nt Ab	7.0 (5.4–8.8) ^1^	6.3 (5.4–8.1)	0.539
RSV/B Nt Ab	6.8 (6.0–9.5)	7.0 (5.4–8.1)	0.859
IgG G	5.0 (3.8–12.5)	5.0 (0.0–20.0)	0.900
IgG F	0.0 (0.0–1.3)	0.0 (0.0–1.3)	0.985
IgG N	0.0 (0.0–5.0)	0.0 (0.0–0.0)	0.088
IgG P	0.0 (0.0–5.0)	0.0 (0.0–5.0)	0.699
IgG M2-1	0.0 (0.0–5.0)	2.5 (0.0–5.0)	0.236
PLA	3.4 (0.5–182.2)	2.3 (0.5–8.8)	0.179
Humoral Ab in convalescent serum			
RSV/A Nt Ab	9.5 (7.4–10.6)	9.5 (7.6–12.3)	0.827
RSV/B Nt Ab	9.5 (7.0–11.0)	11.0 (7.8–13.1)	0.276
IgG G	20.0 (5.0–20.0)	10.0 (5.0–12.5)	0.243
IgG F	0.0 (0.0–10.0)	5.0 (0.0–20.0)	0.511
IgG N	10.0 (0.0–12.5)	0.0 (0.0–20.0)	0.314
IgG P	5.0 (0.0–12.5)	5.0 (0.0–5.0)	0.566
IgG M2-1	5.0 (0.0–20.0)	5.0 (0.0–12.5)	0.612
PLA	7.1 (0.5–768.0)	12.7 (2.1–385.1)	0.751
Mucosal Ab in acute nasal wash			
RSV/A Nt Ab	2.0 (2.0–2.6)	2.0 (2.0–2.3)	0.830
RSV/B Nt Ab	2.0 (2.0–2.6)	2.0 (2.0–3.1)	0.632
IgA G	0.0 (0.0–10.0)	5.0 (0.0–10.0)	0.683
IgA F	0.0 (0.0–5.0)	0.0 (0.0–6.3)	0.974
IgA N	0.0 (0.0–5.0)	0.0 (0.0–5.0)	0.565
IgA P	0.0 (0.0–5.0)	0.0 (0.0–5.0)	0.809
IgA M2-1	0.0 (0.0–0.0)	0.0 (0.0–0.0)	0.366
PLA	0.5 (0.5–0.5)	0.5 (0.5–0.5)	NA
Mucosal Ab in convalescent nasal wash			
RSV/A Nt Ab	2.0 (2.0–2.1)	2.0 (2.0–2.0)	0.334
RSV/B Nt Ab	2.0 (2.0–2.3)	2.0 (2.0–2.5)	0.911
IgA G	12.5 (0.0–40.0)	7.5 (5.0–22.5)	0.770
IgA F	0.0 (0.0–5.0)	5.0 (0.0–12.5)	0.227
IgA N	5.0 (0.0–10.0)	0.0 (0.0–10.0)	0.725
IgA P	5.0 (0.0–5.0)	0.0 (0.0–5.0)	0.430
IgA M2-1	0.0 (0.0–5.0)	0.0 (0.0–0.0)	0.073
PLA	0.5 (0.5–0.5)	0.5 (0.5–0.5)	NA

^1^ Median (IQR) of log2 titers and band intensity scores were used for RSV Nt Ab and WB Ab; Median (Range) of µg/mL was used for PLA. ^2^ Mann–Whitney U-test was used for difference in mean ranks of RSV Nt Ab titers and WB band density scores; Two-sample *t*-test was used for difference in means of log transformed PLA concentrations. IQR = interquartile range; NA = Not applicable. *p* < 0.5 was significant. The serum RSV Nt Ab and serum PLA data were published as geometric mean (95% confidence interval) in our previous publications [27].

## Data Availability

The data presented in this study are available on request from the corresponding author.

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
