# Peer review of "Humoral and Mucosal Antibody Response to RSV Structural Proteins in RSV-Infected Adult Hematopoietic Cell Transplant (HCT) Recipients"

_viruses, 2021, doi:10.3390/v13060991_

Round 1

Reviewer 1 Report

  1. In Table 1, the data were presented with “Median (Range) of log2 titers, band intensity scores and µg/mL were used for RSV Nt Ab, WB Ab…”. However, in Table 2 and Table 4, the data were presented with “Median (IQR) of log2 titers and band intensity scores were shown for RSV Nt Ab and WB Ab”. Why are they inconsistent?
  2. In Table 1, paired t-test was used to analyze the difference in means of the WB band density scores. The WB band density scores were ordinal data and ordinal data has no normal distribution. Paired t-test is not appropriate to use to analyze ordinal data. Can the authors apply an appropriate method to analyze the WB data?

Reviewer 2 Report

The introduction needs some review of humoral vs mucosal antibody response in general and with respect to RSV.

The authors use Western blot analysis to determine what anti-RSV IgA/IgG antibodies are present in the mucosa/serum.  It is possible that this approach may not reveal all the antibodies present. Some antibodies that may perform poorly in Western blots analysis may still be effective at neutrilizing the virus, i.e. conformationally sensitive antibodies. Furthermore, the relative signal on Western blots is not a good indicator of quantity of antibody as nothing is known about the affinity of the antibodies relative to each other. Something should be done to address this problem.

The fact that the microneutrilization assay used does not detect activity from the G glycoprotein and only the F is a real limitation of this work as there appears to be no IgA generated to F but only G in the mucosa.

Round 2

Reviewer 2 Report

Authors should discuss limitations of assays used to obtain the data on the presence of variious antibodies in the mucusal and humoral samples analysed in the discussion.

Author Response

Thank you for the comment.

The following limitations have been added prior to the last sentence of the second complete paragraph in the DISCUSSION (page 10) in the revised manuscript: “The Western blot assay used for detecting the antibodies primarily bind linear epitopes on the RSV structural proteins and may not detect the conformationally sensitive anti-RSV antibodies present in the serum and nasal wash samples. However, we have previously reported on F-specific conformation-dependent competitive antibodies in this HCT population [25-27]. The microneutralization assay used a continuous cell line as its matrix to primarily detect neutralizing antibody activities generated by the F instead of the G protein”.